# Place Attachment and Its Consequence for Landscape-Scale Management and Readiness to Participate: Social Network Complexity in the Post-Soviet Rural Context of Latvia and Estonia

**Joanna T. Storie** [1,*] , **Enri Uusna** [1], **Zane Eglāja** [2], **Teele Laur** [1], **Mart Külvik** [1], **Monika Suškevičs** [1] **and Simon Bell** [1]

1 Department of Landscape Architecture, Estonian University of Life Sciences, Kreutzwaldi 56/3, 51006 Tartu, Estonia
2 Department of Geography, University of Latvia, 19 Raina Blvd., LV 1586 Riga, Latvia
* Correspondence: joanna.storie@gmail.com; Tel.: +371-2-7872380

**Abstract:** This paper uses the tripartite place attachment framework to examine six rural parishes across Estonia and Latvia. Existing analyses/frameworks on participatory processes often neglect the complexity of relationships that rural residents have to their local environments. From a qualitative analysis of face-to-face, semi-structured interviews with case study area inhabitants (23 interviews in Estonia and 27 in Latvia), we depict varying degrees of attachment of individuals to each other and to the place in which they live and their readiness to participate in terms of willingness and ability to participate in a landscape-scale management process. Attachment to the local area was strongest where the social ties were strongest, independent of their sociogeographical features. Social ties were strong where there were good family connections or strong religious or cultural institutions. Taking individual parishes and engaging inhabitants through in-depth interviews using place attachment analysis gives an overall perspective of life in that rural location. These findings reveal important connections within the communities with the potential for planners to engage with local inhabitants and possible barriers to participation.

**Keywords:** qualitative social network analysis; participatory landscape processes; tripartite framework; social ties; barriers to engagement

---

## 1. Introduction

### 1.1. Background

Estonia and Latvia, two of the Baltic States, have undergone significant, intense social transformations. The greatest of these was experienced during the years of Soviet domination between 1944 and 1991 that impacted rural community structure and management [1–3]. The small, independent farms in the countryside were reorganised into collective or state farms, (*kolkhoz* or *sovkhoz*) and landowners were forced to give up their land to become employees with regular hours and social benefits. Agricultural engineers were trained to manage the resulting large units that encompassed many farmsteads. The structure of the landscape was substantially changed as wetlands were drained, new road infrastructure built and urban-style housing constructed in new village centres, complete with shops and administrative/cultural houses [2,4]. On regaining independence in 1991, another social transformation took place as the collective farms were quickly dismantled, causing mass rural unemployment and depopulation. Many of the large farm buildings became redundant, littering the

post-Soviet countryside with derelict buildings [2,3]. Previous landowners or their descendants were allowed to reclaim their land, however, they often lacked the skills or investment needed to modernise them [5–7].

Landscape regulation aimed at preventing unacceptable changes or mediating conflict has a long history [8]. Post-Second World War, the new Soviet state provided everything through centralised land planning policies [9]. This modernist approach regarded the world as a predictable machine through the application of scientific methods [10]. This was a strong feature of the system into which Estonia and Latvia were incorporated, where Soviet ideology sought to bend nature to its bidding [3,11,12]. This reductionist thinking did not envisage people being attached to places amidst a "complex web of non-linear relationships and multiple interactions" [13] (p. 170) or crossing existing geographical boundaries of rural landscapes [14].

Kyle et al. [15] proposed that planners move beyond the modernist idea that landscapes are a collection of attributes to be manipulated to one that was sensitive to inhabitants' relationship to place. Likewise, Dorning et al. [16] argued there is a need to integrate human perceptions and decision-making processes regarding landscapes, to better understand land change processes in order to support landscape planning. Various conventions and policies, such as the Aarhus Convention and the European Landscape Convention, aimed to close the distance between decision-makers and inhabitants who live and work within the landscape by developing new policies encouraging stakeholder participation within a flexible framework [17–19]. Participatory processes, however, are not a panacea for successful stakeholder integration in development. Failure to identify relevant stakeholders or consider their attachment to place glosses over sociopolitical contexts and risks the unidentified stakeholders negatively impacting a project in the future [14,20,21].

It is widely recognised that administrations need to move from government to governance [9] to improve local knowledge integration, local population empowerment, improved process legitimacy and increased trust [20,22,23]. However, countries, such as Estonia and Latvia, which are transitioning from a strong centralised system with modernist paradigms, often lack the political will, resources or skills to move to an enabling role. This contributes to failures in collaborative policymaking or results in ineffective policies [9,10,23].

In addition, alienated, distrustful and deprived communities often lack the confidence to participate; they also lack the political or socioeconomic infrastructure and knowledge base required to organise representative citizen groups [23–25]. A prerequisite for successful landscape management is the ability of the local population to be able to express their opinions and perspectives about their local landscapes [26]. Previous poorly implemented participatory approaches and historical tensions also raise barriers to development [9,25,27], requiring a need for flexible policies and the development of participatory tools [18,23]. The political nature of landscape, e.g., the multiple stakes, power relations and conflicts associated with it, necessitates that these aspects are made more explicit in practical landscape planning [28].

Participatory processes require an understanding of the narratives people build. These narratives influence how people perceive their landscape, how they believe it should be managed and their willingness to engage in planning processes [14,16]. Policies, therefore, need to incorporate residents' stories, emotional bonds formed by traditions, sensory experiences and memories to set the place attachment in context, so that inhabitants feel listened to [29–32]. As Scannell and Gifford [33] explain, memories are the most commonly expressed benefits connected to place, thus it is important that these are strengthened. As the roots developed during childhood are long-lasting and shape adult identity [32], it could be argued children should be included in development planning too.

Landscape-scale management activities may threaten emotional bonds to a place depending on how the process is perceived and there is, therefore, a need to understand the landscape from a human perspective [34]. As Butler [35] states, planners often miss the subjective, intimate, lived-experience of inhabitants. Values that are disregarded or missed in the early planning stages are rarely incorporated later in the decision-making process, so it is important they are included at the start [35]. Poorly

applied management processes can generate negative reactions, where people experience deep distress as either nostalgia, a longing for a cultural setting in the past where a person felt at home, or solastalgia, a period of mourning where an individual feels isolated in their own home environment, or both [36]. As Scannell and Gifford argue, these "broken or 'stretched' place bonds" can cause issues with physical health [33] (p. 256). Thus, successful integration of relevant stakeholders that integrates community values sensitive to their local needs is required in a trusting environment [21,30,37].

Developing collaborative landscape management in rural Latvia and Estonia requires understanding

- rural communities, their history and the issues they consider important,
- the networks, embedded in heterogeneous communities with diverse views and vested interests, and
- an individual's degree of attachment to the physical environment and the community context that influences the intensity of association and how they participate [13,14,21].

These transformations have had an impact on how rural residents relate to place and their attachment to it [38]. Place attachment refers to the emotional bond that connects people to a place and that develops over time as people interact with the landscape [39]. Landscapes become associated with memories that support a connection to the past or to a community, creating a sense of belonging and anchoring people to that place. However, in Estonia and Latvia the authors suggest that, as a result of the social transformations, such place attachment is not strong, leading to weak communities and a lack of interest in planning the future of rural landscapes. Given that public participation in landscape management decisions is one of the key aims the European Landscape Convention, as well as national policy, a better understanding of how the bonds of people to place and how community landscape values develop can be helpful for effective policymaking [40].

### 1.2. Theoretical Framework

Place attachment is not often fixed to one place and people may have multiple places of belonging [31,32]. Internalised landscape values, set within the context of family and culture and the place itself, begin to be shaped in childhood. Multiple anchors are formed [31], inextricably linked with love, grief, pleasure and security, contributing to an individual's identity, "This is what shaped me. It was shaping my identity. It was shaping who I was, who I am" [32] (p. 17).

The place-based values formed over time are important in guiding environmental evaluations that lead individuals to attach to or reject a place later in life and influence their choice of residence [32,33]. As mobility patterns change, people demonstrate varying adaptability to new environments depending on their ability to maintain a sense of continuity or by putting down anchors in a new place or situation [29,38]. Place attachment is also influenced by the mobility of others in a person's social circle or the degree to which a place meets their needs at a particular life-stage [36,41].

Healthy, confident and open communities need bridging, bonding and linking social capital [42]. Bridging social capital consists of the bonds to groups with differing interests across communities. Bonding social capital reflects the ties to close friends and family or with people of similar interests. Linking social capital is the engagement with external agencies to influence policies or draw in resources. In Estonia and Latvia these bonds and ties were weakened or lost as a result of the mistrust built during the Soviet era and the post-independence chaos, where some people benefitted more than others [12].

Place attachment is generally stronger in rural than in urban areas because rural inhabitants choose their home or decide to stay based on factors other than the proximity to services and access to employment [43]. As Stenseke [21] pointed out, this does not mean they make a homogeneous unit. Rural place attachment may reflect links to a person's ancestors or culture; these links strengthen social capital and a sense of belonging [33,44]. Painful memories also affect the range and quality of place attachment, so that personal histories within a landscape can be important [29,33,44]. As well as shaping identity, place attachment can foster a sense of well-being [33,43]. Estonian and Latvian

place attachment suffered disruption, as the identity of rural people, closely connected to nature and as stewards of their own land, was forcibly overlain by Soviet ideology [12].

### 1.3. Research Aims and Objectives

Following the arguments outlined above this paper therefore investigates the role of place attachment among the residents of six rural communities, three in Estonia and three in Latvia. It seeks to answer the following questions:

- How do inhabitants, past and present, experience place attachment, the emotional and relational ties, to the place where they live or grew up?
- What are the push and pull factors affecting continued residence in these rural locations?
- What are the important place-related landscape features and values according to inhabitants, and how can these be taken into consideration in landscape-scale management decisions?
- How ready are the communities to participate in (rural/spatial) development?

## 2. Materials and Methods

### 2.1. Analytical Framework

While past research has often outlined the rules governing landscape development, they rarely describe the landscape and its construction over time that captures the dynamic changing nature of place attachment [29]. Studies of place attachment have focused on the people-people relationships, ignoring people-place relationships and how place attachment develops [44]. Place attachment, along with place identity and dependence, feeds into this sense of place providing anchors to a place [44–46].

Narratives are important in understanding development of people-place attachment and how that is interpreted by individuals [29,47]. Qualitative investigations are therefore needed to understand the nuances of the different patterns that shape place attachment across time and aid understanding of responses to proposed changes [29,33].

Scannell and Gifford's tripartite framework of place attachment [48] supports the investigation of the beliefs and values embodied in place attachment and the processes by which they are acquired. It captures the place attachment values embedded in the social networks of a place, from the individual networks to the wider networks. It allows greater attention to be given to the memories connecting people to a place, because these are important to place attachment and well-being [33]. Through questions based on the framework, the cultural and political influences can be elucidated and a narrative of place constructed. It can also be used to determine people's willingness to engage in management decisions regarding their landscapes and participation barriers.

The analysis of the results was developed from the themes that emerged from the interviews based on Scannell and Gifford's tripartite framework.

### 2.2. Case Study Areas

Three rural parishes in Estonia (vald[1]) and three in Latvia (pagasts[2]) were chosen as case studies (see Figure 1). The rural parish is the level that most inhabitants associate with. The aim of the study was to compare and contrast parishes with good access to markets and other resources with remoter, marginal ones. Defining rurality is fluid, as parishes close to cities have a primarily dormitory function and residents are urbanised as well as having higher tax revenues [49]. Road infrastructure is also important for giving access to other resources. Latvia has been slow to utilise European Union road

---

[1]　Vald (pl. vallad) is the smallest administrative subunit in Estonia. Many of the smaller vallad have now been amalgamated into larger subunits. In this paper the original smaller unit was examined.

[2]　Pagasts (pl. pagasti) was the smallest administrative subunit in Latvia. Administrative reform in 2009 merged pagasti into larger subunits called Novadi (sing. Novads).

infrastructure funding; consequently, minor rural roads are in a poor state. To avoid these urban and infrastructure influences, parishes more than 60 km from the nearest large town and either on a main inter-city route or in a peripheral area were chosen.

The Estonian case study parishes, Adavere and Lustivere, lie in the same municipality, close to the main road between Tallinn, the Estonian capital, and Tartu, the second-largest city. Each parish shows commonalities and differences within a single municipality and similar logistical conditions. Similarly, two corresponding Latvian parishes with similar geographic positions were selected for comparison. Tūja and Svētciems, in the same municipality, are situated on the highway linking the Latvian capital, Rīga with Tallinn. Both peripheral parishes were chosen for their strong and distinctive religious and cultural aspects, contrasting with the country's main culture and being close to border areas. Obinitsa, in Estonia is populated by the Seto minority and close to the Russian border, has a strong Orthodox tradition, and Dagda, in the Latgalian region of Latvia close to the Belarussian and Russian borders, has a strong Catholic tradition.

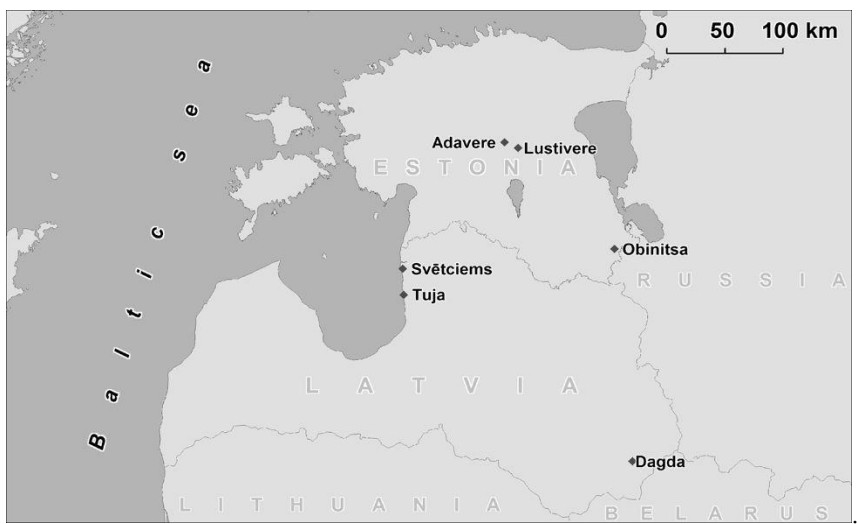

**Figure 1.** Location of the study areas.

The parishes varied in population from 135 to 2178, with limited employment opportunities and limited resources (see Table 1).

**Table 1.** An outline of resources in each parish.

| Area | Population | School | Public Resources | Commercial Facilities | Main Employment |
|---|---|---|---|---|---|
| Estonia | | | | | |
| Adavere | 567 | Primary school kindergarten | Cultural centre, library sports centre, doctor | Small grocery store, petrol station, chemists | School, grocery store, agricultural work, most work in Põltsamaa (the nearest large town) |
| Lustivere | 444 | Primary school, kindergarten | Cultural centre, school has a gym and stadium | Small grocery store, visiting hairdresser | School, grocery store, local agricultural enterprise, most work in Põltsamaa |
| Obinitsa | 135 | Kindergarten | Cultural centre, museum, gym, library, ATM, local church, doctor | Post office, hairdresser, small grocery store, bakery | Stores, local agriculture |
| Latvia | | | | | |
| Tūja | 298 | None | Library, meeting hall | 2 shops, café | Café, shops |
| Svētciems | 379 | Kindergarten | Library with meeting hall | Furniture shop | Shops, most work in Salacgrīva (the nearest large town) |
| Dagda | 2178 | 7–19 years | Municipality office, library, meeting hall/cultural centre, doctor | 3 supermarkets, café (summer only), chemists, 2 ATMS | Municipality, school, shops |

Landscape management in Estonia and Latvia was subject to Soviet rules from the time of occupation until independence was regained in 1991. Since then, the countries have followed similar pathways as they adopted policies in line with the European Union, which they joined in 2004. Both countries have also ratified the European Landscape Convention, however Latvia ratified the convention in 2007 and Estonia in 2018. While Estonia agreed in principle with the convention, there were issues encountered in terminology due to the fact that "landscape" proved difficult to translate into Estonian. The strategic approach of the European Landscape Convention has impacted the landscape planning in both countries with an influence on sustainability planning, and protection of biodiversity and cultural heritage. Although Estonia and Latvia have both formally recognised processes to involve the public in planning, many changes occur that do not fall within the spatial planning system, such as forest management, land management changes, and abandoment etc. Hence, the impact is fragmented at a sectoral level and policies are not well defined regarding participatory processes [50].

*2.3. Research Methods*

Using Scannell and Gifford's tripartite framework of place attachment [48], the use of semi-structured interviews was selected as the most appropriate method for the research. Small populations in each study parish (see Table 1) meant that stratified sampling was not possible, so the aim was to gain a wide range of opinions from as many people who were willing to participate. The questions were open-ended and interviewees encouraged to explain their answers; this allowed the interviewees to talk freely, revealing significant place associations [51]. Figure 2 shows the framework and the questions posed relevant to each element.

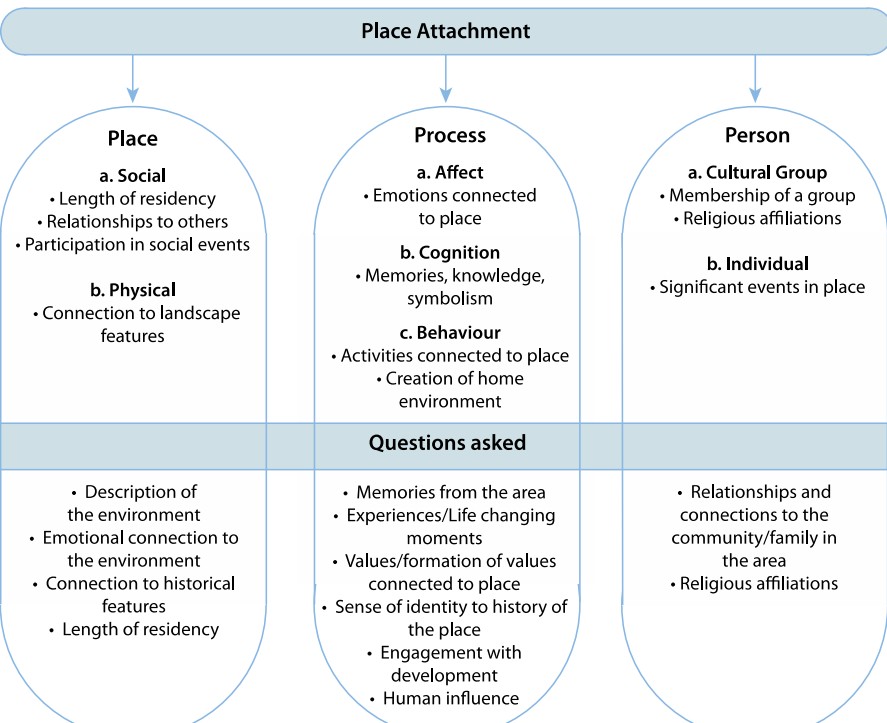

**Figure 2.** The interview questions based on the tripartite framework (source: the authors, based on Scannell and Gifford [48]).

Since Estonians and Latvians consider themselves introverts, building trust with outsiders is an issue. To overcome this, "gatekeepers" were identified through authors' existing networks that provided access to initial interviewees; this was followed by the snowball method, where each

interviewee suggested other people worth speaking to. As Palang et al. [11] and Storie and Bell [25] note, the snowball method is justified in post-Soviet Estonia and Latvia in order to gain trust, but care was taken to avoid network homogeneity [20]. Interviewers also sought out local inhabitants who are in contact with a wide variety of inhabitants, such as activists, librarians, shopworkers and school staff to gain insight into the wider population.

Interview data was collected between 2015 and 2017 by Estonian and Latvian research assistants. Twenty-three face-to-face interviews were conducted in Estonia (Lustivere—6, Adavere—7, Obinitsa—24). Thirty-four individual interviews took place while one interview in Obinitsa was conducted with a group of three people. Twenty-seven face-to-face interviews were conducted in Latvia (Dagda—14, Tūja—6, Svētciems—7). Twenty-six were interviews with a single person and one interview in Dagda was conducted with a group of two. All interviews were conducted in Estonian or Latvian and each was recorded, except for two interviews where detailed notes were made instead.

*2.4. Data Analysis*

The interviews were transcribed and each coded using Qualitative Data Analysis (QDA) Miner LITE software [52] or Text Analysis Markup System (TAMS) Analyzer for Macintosh OSX [53]. The codes used in the transcripts were grouped according to the themes that emerged. All analysis was conducted in the relevant native language and translated into English.

**3. Results**

The results from each parish demonstrated a unique combination of characteristics that determined how well the community functioned and their capacity to develop. Some parishes in close proximity demonstrated striking differences, potentially influencing how landscape planning is presented to the community and the ease of integrating local stakeholders in the process. A short introduction to each parish, elicited from the interviews, is followed by an analysis comparing the spheres of society and external influences that play an important role in place attachment (Table 2 gives an overview of the sections and subsections).

**Table 2.** Details of the following results sections and subsections.

| Section | Subsection |
|---|---|
| 3.1 Introduction | 3.1.1. Lustivere |
| | 3.1.2. Adavere |
| | 3.1.3 Obinitsa |
| | 3.1.4 Tūja |
| | 3.1.5 Svētciems |
| | 3.1.6 Dagda |
| 3.2 Spheres within society | 3.2.1. Individual |
| | 3.2.2. Family |
| | 3.2.3. Wider Community |
| | 3.2.4 Community cohesion |
| 3.3. Influences on place attachment | 3.3.1 Culture |
| | 3.3.2 Religion |
| | 3.3.3 Characteristics of Place |
| | 3.3.4 Sense of home and childhood memories |
| | 3.3.5 The importance of location |
| | 3.3.6 Participation in local community politics |

*3.1. Introduction*

Many of the interviewees grew up under the Soviet system and experienced first-hand the intense social transformations. The following short sections are an overview of the key aspects identified by the interviewees.

3.1.1. Lustivere

Lustivere is situated in a strategic location close to the Tartu–Tallinn road, providing a quiet environment away from heavy traffic but with convenient access to cities. Inhabitants were proud of the 2013 Estonian Village of the Year award. The parish has a low resident turnover and few available apartments for new residents. Inhabitants, however, are increasingly working away from home; therefore, they are not available for community events.

3.1.2. Adavere

Adavere is also considered a strategic location due to its position on the Tartu-Tallinn road; however, some interviewees felt the road dissected the village centre. Inhabitants rarely related to issues outside their own sphere of interest and there was little interest in landscape planning at a larger scale. Interviews revealed distrust at multiple levels and an incohesive community.

3.1.3. Obinitsa

Obinitsa is situated in southeast Estonia and includes a Seto community, an indigenous ethnic and linguistic group with a strong cultural identity. The identity was maintained throughout the oppressive Soviet era and subsequent border issues. There are some current internal issues with a group "holding onto power" from the Soviet era.

3.1.4. Tūja

Tūja is a coastal parish situated on the main road between Riga, the Latvian capital and Tallinn, the Estonian capital. It is mainly dependent on summer visitors for its economy and has a large second-home ownership.

3.1.5. Svētciems

Svētciems is a coastal parish situated on the main Riga/Tallinn road. Many inhabitants formerly worked at the nearby fishing kolkhoz in Salacgrīva during the Soviet era. It is a close-knit community where neighbours consider each other friends.

3.1.6. Dagda

Dagda is situated in southeast Latvia. Its population increased during the Soviet era when several factories opened, but these have since closed. Family relationships were strong with many former residents returning for family events and holidays. The municipality encouraged young families to stay in Dagda by filling positions with local people where possible.

*3.2. Spheres Within Society*

3.2.1. Individual

Generally, interviewees valued the benefits of rural lifestyles (see Table 3). They valued the peace away from the city rush and the space and freedom to pursue rural activities like growing vegetables. They felt out of place in the busy cities. One interviewee considered himself richer than his friends overseas with larger incomes, because he was his own master and life abroad was costly.

**Table 3.** Sense of rural identity.

| | **Rural Identity** | |
|---|---|---|
| Lustivere | I am completely a rural person. We grow cucumbers, cabbages, potatoes by ourselves. I could not imagine things otherwise (Male 58). | I am a rural person. Perhaps with having children I have realized that rural life has more values, but I have never had the need to make a popular decision to "go rural" since I have always been a rural type (Female 36). |
| Adavere | In the city you bump into people. I am shoved and pushed more during one day than I am here for a whole year. I have freedom here. (Female 78). | Generally, I am not such a . . . crowd person. I like to do things alone or things where I have my friends by my side (Female 62). |
| Obinitsa | When I go to the city, I always want to come back as soon as possible. The city life is horrible (Male 64). | When I go to the city sometimes, I get tired fast and I am much more calm here. I would not ever want to go and live there (Female 30). |
| Tūja | In a city I am only a guest for a few days, I like living here, I like the people here, nature, and I wouldn't want to live in a city (Female 51). | I don't like Riga; it is just for work (Male 45). |
| Svētciems | I always would want to live close to sea, close to nature. I don't like big city's rush and noise (Female 34). | I have not lived in a big city, and I wouldn't want to (Male 47). |
| Dagda | I wouldn't want to live in the city with all the noise and the rush and the public transport (Female 26). | I don't like big cities; I have tried to live there but if you don't like them you cannot maintain yourself there (Male 28). |

Many former rural inhabitants wished to return with their families. They felt children had freedom growing up in the countryside. Some, however, preferred the rhythm of city life during the week and the countryside at weekends. Adavere and Obinitsa inhabitants tended to value the private aspect of rural environments. In Adavere, many preferred to keep their distance rather than joining community activities. Younger inhabitants in Dagda and Svētciems, however, wanted to leave, as they felt the countryside was boring.

Individuals displayed resilience and adaptability to cope with the limited employment opportunities. For example, some developed self-employment opportunities in tourism in Tūja and Obinitsa; one Dagda inhabitant combined business consultancy by selling real estate and managing his parents' farm. In Dagda inhabitants displayed adaptability by taking employment offered through their local networks. One graduate in languages, working for the local Dagda paper, explained,

*It is not really the job I want to do myself, but it is a very small town and because I do want to live here in my native hometown, I should take all the possibilities I can for a job. (Female 26)*

3.2.2. Family

Family relationships were important for strengthening place attachment and providing supportive networks. In Obinitsa and Dagda many people returned for family reunions and events, such as Jāņi, the mid-summer festival in Dagda, and cemetery celebrations like Paasapäiv in Obinitsa. Lustivere and Svētciems interviewees expressed a good connection to both family and neighbours.

Family ties within the community in Adavere were weaker, with many people moving to the prestigious sovkhoz during the Soviet period or, more recently, into the area for cheaper housing. In Dagda family ties remained strong despite an expansion in the 1960s with incomers to the new factories; this may reflect the strong Catholic background in the area.

Family roots created a feeling of home, and many interviewees had fond childhood memories. These roots encouraged circular migration, as some took temporary employment, arranged through social networks, to overcome financial difficulties or for education but returned later.

*I moved to Ireland for three years and I came back because I missed home. It is easier to fight the struggles over here. (Female 26)*

In Latvia and Estonia some families became separated after the Soviet Union collapsed, as borders were created that were once porous. For example, the Seto people are divided by the Russian/Estonian border; therefore, easy access to the major trade and ancestral centre was severed. One resident explained,

*Our centre used to be Pechory and ... even if not on a daily basis, on weekends, almost everyone would have some business there. You would go to the church or the marketplace. People who engaged in agriculture could sell their products there. (Male 51, Obinitsa)*

### 3.2.3. Wider Community

Place-based relationships anchor people to an area and interviewees highlighted the advantages of living in a village where people know each other. This contrasted with city life where it is possible to disappear.

*The Estonian is a recluse, does not want to communicate that much, but this is different living in a village. Because in the village we know which porridge is on the table in another farm, and if they have troubles or worries, we would know. (Female 82, Obinitsa)*

In Lustivere, Obinitsa and Dagda, many wished to contribute to the positive atmosphere by participating in community or family activities. However, involvement was limited due to work and family commitments, limited local educational opportunities and health issues. In Adavere, opinions were divided; some interviewees enjoyed being part of the community and others felt threatened by local social issues.

*When I think of Adavere, there is a wide range of people who go from place to place. Rootless people who live in these apartments. (Female 62 Adavere)*

Common issues in all parishes were alcoholism, rural depopulation and declining agricultural employment due to increased mechanisation. Fewer workers meant decreasing social circles as inhabitants moved away, leaving many feeling isolated.

*You could see back then, when you looked out of the window, there were lights on in neighbouring households and someone was in there. But now there is silence. You don't hear the cows; this makes me sad. (Male 55, Obinitsa)*

### 3.2.4. Community Cohesion

Social cohesion and the ability to integrate newcomers varied between parishes. For example, despite Lustivere and Adavere's close proximity, they differed markedly in attitudes. Interviewees in Lustivere were open and worked together for the common good, developing pride and trust, spurred on by winning the Village of the Year award and by well-known and approachable local activists.

In Adavere, however, a criminal reputation caused fear and distress.

*There is a local generation of criminals; a few of them even are our neighbours. There has been an increasing rate of delinquents. These people are the cause of fear. There are also some "mean" locals. (Female 56)*

Activists were working hard to encourage greater cohesion in the community and one activist complained:

*I can never respect those who say, "Oh how horrible it is here in Adavere". I mean ... what do you do in order for it not to be horrible? (Female 44)*

However, there was pride in the local Adavere speed skating scene, where local skaters represented Estonia internationally. Some residents were willing to organise and participate in park maintenance but generally, activism was low due to the community's individualistic and private nature.

*I don't think community-wise. I am very individualistic, only what involves myself. (Female 62)*

Obinitsa had a tight-knit society based on a strong Seto identity, but interviewees tended to view local non-Seto people, who had moved to work in the sovkohoz, as lazy and unwilling to participate in community activities. The Seto were cautious and reserved by nature. "Let's see first how things progress" is a common saying. However, they saw their Seto and Estonian identities as intertwining and enhancing each other. "I have always had the feeling that a good Seto is a good Estonian" (Male 55). There was also a desire to pass on the Seto culture across generations.

Tūja experienced issues with seasonal influxes of visitors, contributing to a waste disposal problem and differences between permanent inhabitants, wishing to develop the area, and second-home owners, resulting in a fragmented community. Some interviewees felt that second-home owners had a sense of entitlement that culminated in restricting beach access and limiting bike lane provision.

*They basically buy a house by the seashore and they think that the sea is theirs. (Male 45)*

No cultural, historical or religious groups were reported by residents in Svētciems, but there is a football club where younger people socialise. Svētciems and Dagda interviewees emphasised the friendliness of local people, particularly in Svētciems.

*Here we know each other, we help each other out when it is necessary and when we have the chance. My neighbours and colleagues are my friends. (Female 34)*

Some locals in Svētciems, though, were described as troublesome, "*like guerrillas (partizans), always calling and making suggestions for change*" and not helping with community activities (Male 53).

Dagda inhabitants consisted of various nationalities—Latvian, Belorussian, Russian, and Polish—who all got along well. Some inhabitants retired to Dagda from Rīga, as properties were cheaper. Workforce quality was considered a problem, as many residents were elderly, unqualified, or had social problems, such as alcoholism.

*The social environment is not good here and that is why people go away and it goes in circles. (Male 28)*

*3.3. Influences on Place Attachment*

3.3.1. Culture

Estonian and Latvian identity is strongly tied to the culture of theatre, song and dance with widespread participation in groups, across ages and from local to international levels. Some interviewees believed that cultural activities, particularly in the Estonian parishes and Tūja, brought the community together where supportive group members played an important role.

*Basically it [the theatre group] is about communication. People are coming from different corners of the parish. They are coming with different problems. There are solutions made while people are communicating; there are business relationships made. (Female 54, Tūja)*

However, participation in cultural activities has diminished due to time constraints, ill health, and outmigration with people working abroad or in distant cities.

Although politically controversial, older inhabitants generally had positive memories of cultural activities during the Soviet era. This was evident in Adavere as the esteemed local sovkhoz was well funded and guided by an enthusiastic leader who maintained social and cultural activities within the community.

*I don't want to sing an ode to the Soviet era, but ... the sovkhoz ... they financed culture. It had a culture centre, it had decent workers; they were invited, they had theatre, whatever. It was strongly represented. When it collapsed, there wasn't that inner drive anymore that would've kept it all together. (Female 56)*

However, some had negative recollections:

*When the Soviets came, then things changed. Initially even when they worked in the kolkhoz, they would go and sing after work. You could hear how women sang while returning, they were leelo-ing. Then in the second year all the singing disappeared. Everyone realised what had hit us. (Female 82, Obinitisa)*

When the Soviet regime collapsed, local activists again gained freedom to organise local cultural activities.

*Some make a living out of culture. You don't have anything to fear now. Before you were afraid, because if you emphasised your culture too much, you could've been repressed. Now you don't have to fear that, just do it. (Male 51, Obinitsa)*

In Obinitsa, the Seto Societies united the Seto people through an intensive cultural schedule, including choral singing, dancing, handicraft workshops, theatre plays and film productions. The Seto Society published literature on Seto culture and architecture. The older generations remembered the rituals performed by their ancestors and the land was rich in cultural symbols, such as historic graveyards and weirs (see Table 4). Natural areas often had their own unique tales and legends.

**Table 4.** Obinitsa traditions.

| Obinitsa Traditions | |
|---|---|
| Traditions Connected to Nature | My grandmother used to go to the forest, and she communicated with the forest god. The first berry would always be left for the forest god and the first mushroom (Female 82). |
| | Even nowadays they believe that when you wash your eyes in the stream of Meeksi, it will heal your vision (Female 30). |
| Community Traditions | Interviewee 1: Yes, we have watchful eyes here: "Why are you working on a holiday?" Interviewee 2: It is very important to follow these traditions. They will tell you if you make an error (Female 37). Interviewee 1: But you should not take it as a negative thing. Since the youth have their busy lives, they come and help you. You should not take the scolding seriously. (Group interview with Female 30 [interviewee 1] and Female 30 [interviewee 2]). |
| Architectural Traditions | We used to have those wooden gates, which are now made of iron. Wooden curtain rods have also been replaced with metal. A lot of archaic things have been replaced (Female 52). |
| Borrowed Traditions | The Setu folk have a lot of things borrowed from the Russians. Words, choice of colours, clothing. They are quite close by (Female 37). |

The Lustivere Village Society foundation was considered an important local milestone and a unifying feature for the whole parish. Schools also played an important role in cultural, sporting and outdoor activities. The Lustivere school organised a summer camp promoting environmental care through outdoor work activities. In Dagda, a teacher set up a ping-pong club and in Adavere, the school played a central role in community building through school plays, which is important in a community lacking community cohesion.

3.3.2. Religion

Generally, church influence was low, except in the peripheries; Obinitsa has a strong Russian Orthodox presence and Dagda has a strong Catholic presence. Some locals in Obinitsa observed the

Estonian neo-pagan religion (Maausk) with local sacred places, such as groves, trees and boulders. In Obinitsa, many gathered for the frequent church events demonstrating the corporate nature of religious observance in this area. This contrasted with the personal nature of religious observance in Dagda, where many interviewees mentioned they attended church on a regular basis.

> *I go for inner peace; basically I simply have to do that because I am a Christian. Not that I really like it or enjoy it, I simply need to. (Male 39, Dagda)*

In Obinitsa church events were a traditional way to socialise, especially the religious holiday Paasapäiv (Transfiguration Day). Families gathered from all over Estonia and Russia to picnic at their ancestral graves to remember and honour them. Despite the Soviet era, the Setos continued to observe Paasapäiv, as the partorg (Communist Party local branch leader during the Soviet era) organised the work schedule around the holiday and planned his vacation accordingly.

### 3.3.3. Characteristics of Place

A quiet natural landscape with fresh air was important to many interviewees (see Table 5). Interviewees described feelings of peace, freedom, happiness and satisfaction. Solitude was also mentioned as a reason for living in Adavere, Obinitsa and Svētciems. One interviewee stated:

> *It is peaceful, it is calm, there is nature and there is everything you need. There is a grocery store, a doctor, there's a pharmacy, and the big city is not for me. (Male 73, Dagda)*

**Table 5.** Connection to a Physical Place.

| | Connection to a Physical Place | | |
|---|---|---|---|
| Lustivere | It is kept tidy. The park is very tidy. The pond has a fish ladder, it is lovely. The human factor. It is beautiful (Female 36). | Clean air, birdsong, peace, quiet . . . which keeps the person healthy (Female 36). | I think Lustivere is a beautiful place. There is a manor park where one can walk which is definitely a plus. There are many pleasant places near Umbusi river and plenty of springs around (Female 60). |
| Adavere | I don't say that I go to the countryside every time. But it is a safe place to retreat to, I know that (Female 56). | I don't know... the peace and quiet. I like when I go to jog in spring and how the first flood water is gurgling (Female 20). | When we go and clean the park with our school staff and students, then the elderly ones always join in (Female 44). |
| Obinitsa | I think this liberty and a greater personal space [connects me to this place]. But then again, I notice that those who come here, don't really engage in [local events as much as] perhaps those who have been here their entire life (Male 55). | I like the varying landscape. Vast plains so you can see far. And then there are those hills and valleys. And there are lakes, river Piusa. The riverbanks are beautiful, those sandstone cliffs. And old water wheel ruins. The sandsea of Piusa is also close by, even though it is artificial. But it is a lovely place (Female 24). | You can do your own thing. There is room to breathe (Male 40). |

**Table 5.** *Cont.*

| | Connection to a Physical Place | | |
|---|---|---|---|
| Tūja | It has been my family's property from somewhere around 1890s, so a long time ago (Male 45). | You can see the beach, the forest, these bicycle lanes and these old roads that do not have any asphalt they go right beside the seashore. So, if anyone wants to see something beautiful, he will because each place has its own beauty (Female 51). | I like the sea, so we decided to build a house here (Female 56). |
| Svētciems | It is hard to say; it is just my home now. I have lived all my life in this area, I always would want to live close to sea, close to nature (Female 34). | As my house is close to the sea, I can say that the place I work, is not so far. Basically, you can say I work at home (Male 72). | I have lived all my life in this area, I always would want to live close to sea, close to nature. [Female 34] |
| Dagda | I want my children to grow up here because it is much better here in Latgale, not so much polluted air (Female 26). | It is a place of many islands there are many important sites in Dagda … it is very special … (Female 57). | I cannot imagine (living) anywhere else, the people, the nature is unique in such a small place. For example when my daughter brought her fiancé here for the first time I told him, "Oh wait I will take you to Dagda's park," and he was amazed and said, "The whole of Dagda is like a park, do you still have a special place called a park?" (Female 64) |

Although many interviewees disliked the large agricultural operations' practices of using chemicals, they preferred the neat and tidy appearance to encroaching scrub. It was considered an improvement on Soviet agricultural practices which led to water pollution and negative impacts on biodiversity and nitrate sensitive areas.

The flat agricultural landscape in Lustivere and Advere with some forest stands was considered unremarkable, but still valued.

*When you think about it, there is not much to say about it. But this "ordinary" landscape has its own charms as well. (Female 50, Lustivere)*

Also,

*When you look at the grain field moving in the wind. Is there anything more beautiful than that? (Female 78, Adavere)*

Some interviewees disliked the Adavere landscape, describing it as "lousy"; some expressed concern about the highway that cut through the village.

*What is a whole different topic is the highway. I don't even want to talk about that. This divides the community and goes right through these beautiful fields here … splits them in half. (Female 44)*

Interviewees appreciated Obinitsa's hilly terrain, as it favoured the more traditional extensive farming, which they preferred.

*The current technology cannot handle the hills and cultivate these. But otherwise … in every possible place they push their ploughs into the soil and start cultivating. Only our unique landscape keeps them at bay. (Male 51)*

Coastal erosion in Tūja was a concern: the local lighthouse, which was 60 metres away from the sea 35 years ago, now stands in the water:

*The sea is coming towards the land. My basement door is 2 metres away from the beach. After the storm, it comes closer and closer. (Female 63)*

Inhabitants of both countries joined in national environmental clean-up campaigns at the local level, but littering was still considered a problem with annual spring-clean campaigns having limited effect in some places.

Abandoned land and decaying buildings from the Soviet era were common features throughout Estonia and Latvia as people migrated to the towns or abroad after the restoration of independence. Recently, foreign-owned companies have purchased land and larger-scale farmers rent extra hectares from local people, resulting in limited land for farmers wishing to expand. Even though interviewees in Svētciems described some issues with dangerous buildings, abandoned buildings were not generally considered an issue. When people migrated, they retained ownership of rural properties or inherited them from relatives. These properties were maintained but not occupied year-round. Abandonment was an issue in the more sparsely populated border areas.

There were few environmental problems due to limited industry in the case study areas. The relatively unpolluted landscape meant that many interviewees could enjoy many rural activities, such as mushroom and berry picking, gardening, making medicinal herb teas, and fishing. However, felling in the forests caused distress to local inhabitants demonstrating a lack of cooperation between the state forest management and local inhabitants or participation in the decision-making process:

*We get confused when we go to pick berries and mushrooms and there is no forest any more.*

*(Female 53, Dagda)*

Some respondents raised issues regarding regulations within Nature Protection Areas; for instance, one interviewee explained that he was prohibited from cutting firewood, as it would disturb the insects in the ecosystem.

*Where my forest ends is the State forest and the State forest is cut down to the ground. (laughing) and the bugs do not go any further right? (Male 45, Tūja)*

Although generally there are few environmental problems, a leak from a Belarusian chemical factory upstream from Dagda killed fish in the local river, and there is an ongoing eradication programme for the dangerous, invasive Siberian hogweed introduced as animal feed during the Soviet era.

### 3.3.4. Sense of Home and Childhood Memories

The interviewees' sense of home generally centred on relationships or an attachment to rurality, such as the space to grow their own food. Sometimes there was attachment to local features, such as the Obinitsa valleys, the sea by Tūja and Svētciems and the lakes around Dagda.

*Oh, yes, I have lived here all my life. I have been a fisherman all my life. As my house is close to the sea, I can say that the place I work is not so far. Basically, you can say I work at home. (Male 70, Svētciems)*

The sense of home was unique to each individual, for some a refuge in times of crisis and for some a source of pride. Obinitsa interviewees had the strongest attachment, which was reflected in their sense of history that had evolved over a millennium. Their culture is also deeply rooted in the unique hilly terrain of southeast Estonia, with dense pine forests.

*The hills dominate, and from whichever side you approach Obinitsa, you have to cross a valley, a deep valley. These are exciting valleys; the locals always have some memories connected to them. (Male, 51)*

Many interviewees had fond childhood memories of natural features such as trees, the agricultural chores, experiencing the rhythms of nature and the freedom of playing in a natural environment (see Table 6).

**Table 6.** Childhood memories.

| **Childhood Memories** | | |
|---|---|---|
| Lustivere | I used to go and see where the first primulas would bloom, where they used to be. I go to see different flowers. I know where they bloom during spring (Female 36). | In my youth I used to herd cows and walk through all the fields with my herd. And those riverbanks where I played with the flowers. All of my youth plays a part. It is probably the reason why I feel connected to this environment (Female, 60). |
| Adavere | I am glad that I am from a rural place. I like that I have not grown up in a city. I think it gives the children so much more when they can grow up in the countryside and play in the open and have no boundaries. I remember how we had running and bicycling competitions and played near the fields. I have fond memories of my childhood games and I think the countryside is a good place to grow up in (Female 20). | |
| Obinitsa | I remember what was done back then. I remember how we went to the forest when I was 5, I remember how we visited the forestry crew. And how we used to make things of alder bark. And how I rode downhill with my bicycle and fell over. And how the locals were speaking their stories of things that had happened in the village recently (Male 55). | There are fields. I remember in my childhood how we would pick the barley heads when father had harvested and stored them. And we had to pick it all, it was taught very early to us. And how we kept the land when we made hay. It was all quite spiritual (Female 82). |
| Svētciems | I cherish that I have had a chance to live so close to sea and enjoy summers on the beach (Female 23). | Wonderful childhood! The sea, fresh air, nature, peace. Memories (Female 49). |
| Dagda | I have dear memories in the school in Ezerrnieks by the shore it is very beautiful and it is in a park and the school itself is like a historical heritage (Female 53). | I was born here and I couldn't imagine living anywhere else (Female 57). |

### 3.3.5. The Importance of Location

During the Soviet era, Adavere had a sovkhoz, a prestigious state farm, and many inhabitants moved there as farming specialists. Today, its central location close to the main Tallinn-Tartu road allows some flexibility in employment and its rural nature is considered an important attribute by interviewees; attitudes were similar in nearby Lustivere.

Tūja is considered a desirable coastal region close enough to Riga to attract tourists. The beach is an asset, and the local municipality requested a campsite to maintain access to it. Organising facilities such as public toilets and a lifeguard station proved difficult due to a biosphere reserve and the coastline's protected status. Permanent buildings are forbidden within 300 metres of the beach and the campsite owner complained, "Why do you need a lifeguard building in the forest?"

Dagda interviewees complained that restrictions in Rāzna National Park hindered livelihoods. Special permission was required to fell trees or build new houses. However, the park's scientific value attracted researchers and students. Dagda's church is also a cultural heritage object restricting possible changes to windows to combat the winter cold.

### 3.3.6. Participation in Local Community Politics

One might expect that strong place attachment leads to a willingness to work to improve the community through active involvement in local politics. However, participation at the local political level varied between individuals. One interviewee from Tūja area and one from Dagda were deputies,

some were previously deputies, and one from Tūja and one from Svētciems were referred to as elders (elected as liaisons between the local authority and the smaller parishes to represent their interests). Emotions in the community, however, generally ranged from a hopelessness regarding the current situation to a united resolve that could influence decision-making processes.

Some interviewees in Dagda believed the community could not affect the decision-making process, because "no one ever speaks up much" (Male 28), and some in the municipality "do what they want to do and they pretty much don't do what the society needs and wants" (Male 73). Some suggested there was nobody able to "fight for society's needs" (Male 39), as many young adults had left and society now comprised mainly of elderly residents and local government workers.

In Obinitsa, there was a similar view to the local authorities as it was suggested that several local projects were initiated without proper consultation or engagement with the local inhabitants, which resulted in distress.

> *"Participation is more a formality here. In the end, the municipality acts as they see fit". (Male, 49 years)*

Generally, interviewees stated time constraints limited involvement, but some admitted they do not "like things complicated" (Male 72, Svētciems), and, "Maybe I am too lazy to do that kind of stuff" (Male 47, Svētciems). Attending meetings was described as a "headache" (Female 29, Dagda) that only municipality workers attended. However, people were motivated to participate when an issue affected them. Efforts to encourage participation in Dagda through meetings in different parishes were considered a failure as people did not attend, but the elder in Svētciems approached gathering people together in a different manner.

> *I will have an event tomorrow (Easter) at my guesthouse, to motivate people to act. I am going to grill herrings and invite people over like old times. (Male 53, Svētciems)*

## 4. Discussion

The analytical framework proved a useful structure for interviews, generating many insights into the current composition and issues within communities. As Scannell and Gifford [48] (p. 3) explained, "the cultural and individual levels of place attachment are not entirely independent", meaning the analysis is complex, with many overlapping themes. As our research showed, separating the Process aspect from Person and Place was inadequate to describe the complex interrelated processes connecting people to place. The various aspects are more usefully viewed as flowing into each other through a process of people influencing place and vice versa (see Figure 3).

Scannell and Gifford [48] argue place attachment has a multidimensionality that differs in type and level, our results show this to be the case. In general, although there are some similarities between parishes with similar socio-geographical features, a more nuanced picture emerges, embedded in the networks of the heterogeneous communities with their diverse views and vested interests.

Those choosing to stay in rural areas did so because they preferred the peace and disliked the city rush. In Adavere the emphasis was on the privacy, whereas in Dagda, Svētciems and Lustivere, there was emphasis on the positive benefits of a close-knit and friendly community, with Dagda having the greatest emphasis on family ties. Supportive family networks were important in times of crises, demonstrating the importance of these networks. Each aspect acted as anchors to the place (see Figure 4).

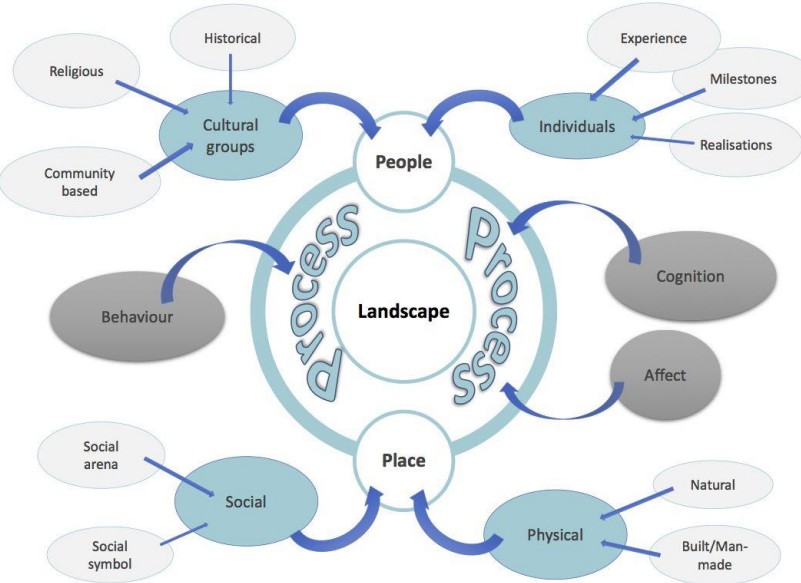

**Figure 3.** A process-oriented model of a Sense of Place, depicting the dynamic processes of the concept (Source: the authors).

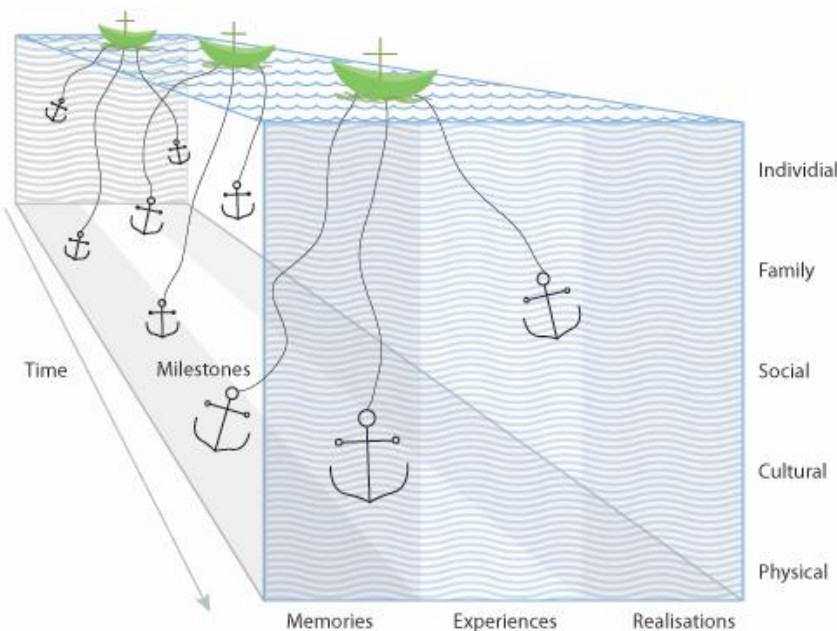

**Figure 4.** Place attachment can be visualised as anchors, with boats representing the individuals' trajectory through life. Each memory, experience and realisation creates an anchor to a place within the context of individual, family, social, cultural and physical interactions with the place. Anchors can be consciously raised allowing people to move on or people experience loss, such as the loss of social connections which may sever the connection to a place, leaving people feeling adrift. This visualisation aims to demonstrate the fluid nature of place attachment, rather than as a fixed entity. (Source: the authors).

Ryan [34] states that the landscape perspective is important and financial considerations are not the only push/pull factors; therefore as Butler [35] insists landscape assessment decisions need to take into account inhabitants' emotions, networks and actions connected with the landscapes. Rural people tend to form communities of place rather than communities of interest (common in urban areas) [48]. As expected, our results showed a strong attachment to the peace of the countryside. Many interviewees describing themselves as "not city folks", choosing to live there despite resource

constraints, up to a point [33,43]; however, financial considerations and family circumstances did take precedence. This has repercussions for the future regarding inhabitant inflow and outflow caused by the limited employment opportunities. Inhabitants may identify themselves to be a rural person independent of a deep connectivity to a particular area but demonstrated settlement identity typical of a mobile population such as Latvian and Estonian post-Soviet rural societies [38,41,44]. Settlement identity is where a person is attached to a settlement type, rather than a specific geographical place.

Many interviewees' current home reminded them of the place where they grew up, or the place to which they returned after time away. There was evidence of repeated residential mobility or circular migration as inhabitants took up short-term work opportunities. Whilst some found temporary migration an enriching experience as they realised the value of home, for some their place attachment had become tenuous and their well-being suffered when economic constraints took priority and social opportunities decreased [33,38].

Many interviewees experienced solastalgia, as the place failed to meet all their needs and as neighbours left the area, creating an increasing sense of isolation. This was exacerbated by the feelings of hopelessness in being able to affect the changes occurring within their landscapes. Although older interviewees did not want to return to the Soviet era, they mourned the loss of work and community associated with those times [36]. Neither did they feel the new governance structures were working for them or with them, for example the lack of consultation on significant landscape changes. Even younger interviewees who grew up after the Soviet era ended were worried for the future of the parishes where they lived as the population declined. Depopulation reduces the strength of attachment, as community connections decreases, weakening the social anchors to place.

Bailey et al. [29] and Scannell and Gifford [33] suggest that place attachment should be explored across a person's lifespan, as previous experiences influence present attachments. Our interviews illustrated the importance of memories to rootedness in a place. Strong attachments were often centred on connections to the community through strong bonding social capital, particularly the family circle, despite family breakdowns and family members working abroad.

The strongest family ties were found in Obinitsa, which reflected attachment to the strong local culture. As Kizos et al. [42] noted, a combination of bonding, bridging and linking social capital leads to a more effective participatory and trusting environment. We found this to be true to some extent but due to the introverted nature of the Latvians and Estonians, stronger family ties were sometimes at the expense of either local ties or external ties. Adavare had low bonding social capital due to individualistic attitudes and ties to family living elsewhere. Conversely, Obinitsa had low bridging and linking social capital due to strong tribal ties (bonding social capital). Both communities experienced higher levels of mistrust that will influence their ability to cooperate in landscape management decision processes in the future.

Activities that increased place attachment included working on common tasks such as park clean-ups or agricultural chores, such as potato harvesting. In Obinitsa and Dagda, local religious institutions and events were influential in place attachment, providing a supportive role to strengthen social capital and identity. Family reunions also strengthened place attachment as the space afforded by the rural homesteads were utilised to reinforce family connections, creating positive memories in a rural context.

Interviewees had varying levels of attachment and of motivation to participate in community events and landscape decision-making; some were, or had been, elected representatives and some were unmotivated and believed it to be a hassle. Readiness to participate appeared to be dependent on individual motivations and perceptions rather than on attachment to place, as many believed their individual voices would be unheard if they did participate. This reflects the Soviet influence on the communities, which many still remember, although even younger interviewees sometimes expressed hopelessness that decision-making rested with an unresponsive government.

The socioeconomic infrastructure and knowledge base required for participation was lacking, and while this is changing there are still significant knowledge gaps in moving from authoritative state to

participatory governance [23]. Top-down authoritarian decision-making—for instance in clear-felling of forests where people collected berries and mushrooms—is evidence of a lack of cooperation between the authorities and the local inhabitants requiring a bottom up social innovation to facilitate trust [23]. As Eiter and Vik [26] also explain, some people do not feel comfortable with the format of public meetings and attendance at meetings can be a problem in rural areas. However, people were motivated when they believed development or planning would affect them personally.

Place attachment and identity is negotiable through socio-cultural and political processes, where landscapes are constructed through both global and local impacts. As Arts et al. [14] (p. 451) argue, "the shaping of landscapes should be the starting point of any governance analysis, and not their fixed state of the art." Governance should therefore start with the landscape values of inhabitants, and this research demonstrates that the application of Scannell and Gifford's tripartite framework of place attachment [48] is capable of revealing values that citizens hold and their roles in the landscape that are needed for more sensitive planning [30,37].

Rogge et al. [9] (p. 335) found "The multitude of actors, the integration of different knowledge systems and the presence of different policy levels can result in a complex interface", which was evident in these six cases. Rogge et al. [9] also explained the necessity to work with multiple stakeholders to manage the complex issues and to find the "building bricks that shape these processes". The analyses summarised here cannot fully address the complexity of each community or the issues they face, but they give a snapshot of community structure and various avenues for working with communities by highlighting opportunities and barriers. The key to integrating multiple stakeholders is through identifying important nodal points where connectivity is highest for effective knowledge transfers. Building these networks and experience of collaboration will lead to a reduction in the time taken in participatory processes in the long term [23]. As highlighted in this study, family connections are strong in some areas and these can be utilised to improve communication, a potential often missed by landscape managers.

It is important to note that the quality and depth of the interviews improved when interviewers had prior connections to the area; however, that restricted the choice of sites. Many people are willing to talk about places they love but high levels of mistrust in Latvia and Estonia often result in a reluctance to talk. One interview was proceeding slowly until the interviewee realised she knew the interviewer's grandmother; her guard then visibly relaxed. Only Svētciems with its more friendly outlook did not require prior networks.

Gatekeepers, therefore, were important for accessing networks; as Stenseke [21] noted, trusted relationships are key for successful outcomes. A powerful position in the community was unnecessary but gatekeepers needed to be well respected. For example, a carer at a local high school, with a motherly nature and well known to teachers, parents and pupils, both past and present, helped in gaining access to inhabitants from a wide age range and background. Still, the authors believe that an influential position could be a hindrance where hostilities exist within communities. Therefore, the snowball technique was helpful, although as Luyet et al. [2012] notes, care had to be taken to prevent network homogeneity being reproduced. Attention was given to using multiple entry points, ensuring that alternative views were taken into account. Shop workers, bus drivers and carers are all key network entry points into rural life.

## 5. Conclusions

In this research we asked four questions:

- How do inhabitants, past and present, experience place attachment, the emotional and relational ties, to the place where they live or grew up?
- What are the push and pull factors affecting continued residence in these rural locations?
- What are the important place-related landscape features and values according to inhabitants and how can these be taken into consideration in landscape-scale management decisions?
- How ready are the communities to participate in (rural/spatial) development?

To answer the first research question, our work has revealed the unique place identities of each small rural community by ensuring a comprehensive overview of place attachment through time and the effects this attachment had on its inhabitants and the landscape. We have shown that people demonstrate strong attachment to the local area where the social ties were strongest, such as good family or neighbour connections or strong religious or cultural institutions, independent of specific sociogeographical features or proximity to resources.

Regarding the second question, even the communities with the pull factors of strong networks demonstrated that place attachment is insufficient to overcome the economic restraints in rural areas. Communities struggle with problems such as restricted access to resources, e.g., education and medical facilities, unemployment and low viability of small farms that often act as push-factors forcing people to move away for work. Thus, communities are not thriving, as evidenced by dwindling populations, and need extra resources to reverse the flow of people away from the rural areas. Even though economic considerations are not the strongest anchors, they are the strongest enablers for the attachment process by providing a livelihood facilitating positive memories and connections to others.

The answers to the third research question generally were not very specific in terms of the important features and values of the physical landscape. The landscape features that interviewees noted were the peace and quiet, in contrast to noisy urban environment. Inhabitants preferred well-maintained areas, such as agricultural fields rather than untended scrub. They also noted a number of environmental problems or concerns which differed from place to place. Access to forest, however, was a valued feature of the landscape and maintaining access for mushroom and berry picking was considered important. There is a need, therefore, for forest managers to work with the local inhabitants to reduce distress caused.

The answer to research question four, on the willingness to participate in landscape development, is that while shared visions and networks exist, particularly through family and cultural connections, they are insufficiently utilised for development purposes. The weaknesses in social capital and the level of interest and willingness to participate in local community politics do not bode well for building participatory processes in these parishes. Building trust in management processes is needed. Our results also indicate issues that need addressing, for example, low community cohesiveness in Adavere and issues between permanent residents and second-home owners in Tūja that require participation from multiple stakeholders within a safe environment with trusted facilitators.

Building a picture of rural life with its networks and connections, where vibrant communities are built and supported, is needed. It should not romanticise hardships but recognise the positive diverse aspects of rural lifestyles [14,21]. These findings can only be considered as a starting point for landscape managers to access valuable information routes and connectivity, which will need constant re-evaluation through the process to remain relevant to the community and ensure social inclusion of relevant stakeholders. This research demonstrated the advantage of building a trusted network (gatekeepers) that would allow meaningful access to communities and build relationships, helping the authorities to move from authoritative governance to participation. This is particularly important in post-Soviet countries with a high degree of mistrust [23] like Estonia and Latvia. Much can be done to help to activate the most marginal communities and the research shows quite strong differences between the sample parishes, where different routes to place attachment can be found. Thus, no one-size-fits-all solution exists, but each place needs to be understood separately and the key factors identified before serious participatory planning can take place.

**Author Contributions:** Conceptualization, J.T.S.; methodology, J.T.S. and M.K.; formal analysis, J.T.S.; investigation, E.U., Z.E., T.L.; data curation, J.T.S.; writing—Original draft preparation, J.T.S., M.K., and M.S.; writing—Review and editing, J.T.S., M.K., M.S. and S.B.

**Funding:** This research received no external funding.

**Acknowledgments:** We would like to acknowledge the help in design of the figures for this paper. We thank Janar Raet for the design of the map for Figure 1 and Gloria Niin for harmonising the designs from the authors' ideas for Figures 2–4.

**Conflicts of Interest:** The authors declare no conflicts of interest.

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
