# Peer review of "Place Attachment and Its Consequence for Landscape-Scale Management and Readiness to Participate: Social Network Complexity in the Post-Soviet Rural Context of Latvia and Estonia"

_land, doi:10.3390/land8080125_

Round 1

Reviewer 1 Report

This paper has some technical merits in its investigation of place attachment, landscape planning and readiness to participate using a case study approach. The paper is theoretically well founded. The manuscript has good potential to attract readership but however does not currently meet the publication standards of Sustainability and needs some revisions.

The authors however need to articulate more clearly how the current study bridges the gaps between theory and practice. The aims and objectives of the research need to be stated at the outset of the paper.

The authors need a literature review section. There are a number of recent work on place attachment e.g. Ramkissoon, Mavondo, & Uysal, 2018 (Journal of Sustainable Tourism); Jiang et al., 2017 (Journal of Hospitality Marketing & Management that may help to enhance the flow of arguments. Authors also need to review literature on community participation – Nunkoo et al. (2012) Annals of Tourism Research and  stakeholder engagement e.g. Sowamber et al. (2018); Nunkoo & Ramkissoon (2017) - Journal of Hospitality & Tourism Research

The methodology is sound. The discussion of findings however needs to be adequately tied to literature. This section needs to be carefully revised and linked to relevant studies. Authors should clearly stress on the selling points on their paper. The conclusion needs to tie to the rest of the manuscript. Overall, this is an interesting manuscript and if carefully revised, it has potential to attract good readership. Thank you, and good luck with the revisions.

Author Response

The article is interesting, especially the results that are clearly presented. But I have some doubts about the methodology and the aim of the research. According to the title, it should be about “place attachment, landscape planning and readiness to participate”. However, it seems to me that is only focused on the first topic. No information about the local landscape planning system and no information about the possible involvement of population are given. It seems to be a research on place attachment, but there is no sufficient information for an efficient landscape planning, since the main collected data refers to the feelings and the emotions of some citizens. In lines 596-599 authors state that “…is capable of revealing the values of the citizens planners need to for (please correct) sensitive planning”. I do not agree with this statement, the values identified talking to some (few) citizens are not enough for landscape or territorial planning. For an adequate landscape planning there is the need to deeply study the landscape structure and transformations, landscape perception, importance of different landscape features, needs and expectations of the local population. 

We thank the reviewer for pointing this out and on reflection the emphasis of the paper as expressed in the title is inaccurate. We have therefore shifted the focus and dropped the references to planning, instead discussing landscape change, management or development, all aspects which have an impact on people.

Since we have done qualitative research we do not aim for representative samples of residents but through the sampling approach we obtained access to a typical range of residents who were willing to be interviewed. The interviews were conducted using gatekeepers and interviewees who had significant contact with other inhabitants. The small sample size reflects the small number of inhabitants of the parishes but as much as possible covered a range of ages and employment. However, the small number of younger inhabitants interviewed reflected the small number of young adults in the rural regions of Estonia and Latvia, as they leave for jobs and further education. 

It was recognised that the snowball technique could lead to issues with homogeneity of the sample and therefore steps were taken to try and encourage contact with those who may have a different opinion. It was also pointed out in the text that this is a starting point for planners to consider alternative networks that existed in the parishes, for example the inclusion of whole families, rather than meeting formats that did not work well. 

Feelings and emotions are an important part of landscape values and especially important to two countries who have experienced their sense of identity being overwhelmed by the Soviet ideology and unable to flourish under its harsh regime. It is our experience that feelings and emotions of the citizens are the primary way that inhabitants of these two nations express their relationship to their environment.

Some examples of studies of the importance of public participation in landscape planning in different countries and frameworks:

Thank you for these suggestions. Those that we could obtain were read with interest and incorporated where appropriate and the paper updated with more recent publications.

Eiter, S.; Vik, M.L. Public participation in landscape planning: Effective methods for implementing the European Landscape Convention in Norway. Land Use Policy 2015, 44, 44–53

Dian, A.M.; Abdullah, N.C. Public participation in heritage sites conservation in Malaysia: issues and challenges. Procedia-Social and Behavioral Sciences 2013, 101, 248-255.

Bruns, D. Cultural Landscape: All That People Give Value to in Their Surroundings. In: Basic and Clinical Environmental Approaches in Landscape Planning. Shimizu H., Murayama A., Eds.; Springer, 2014. pp 3-13.

Paavola, J.; Hubacek, K. Ecosystem services, governance, and stakeholder participation: an introduction. Ecology and Society 2013, 18(4), 42.

Prieur, M.; Durousseau, S. Landscape and public participation. In: Landscape and Sustainable Development. Challenges of the European Landscape Convention. AA.VV.; Council of Europe Publishing: Strasbourg, France, 2006.

UNECE United Nations Economic Commission for Europe. Aarhus convention. 1998. www.unece.org/env/pp/treatytext.htm

Selman, P. Community participation in the planning and management of cultural landscapes. Journal of Environmental Planning and Management 2004, 47, 365–392

Stenseke, M. Local participation in cultural landscape maintenance: Lessons from Sweden. Land Use Policy 2009, 26, 214–223.

Ryan, R.L. The social landscape of planning: Integrating social and perceptual research with spatial planning information. Landscape and Urban Planning 2011, 100, 361–363.

Here some punctual observations:

54. Specify briefly the “various conventions and policies”. At least name the European Landscape Convention, for sure the most important in the last decades.

Added

126. I would not use the word “choose”. Home is not always a choice, but in most of the cases it is simply the place where you are born or where you grew up.

This is an interesting point but many people choose to stay or leave the place where they grew up. We have clarified this by amending the text to “Place attachment is generally stronger in rural than in urban areas because rural inhabitants choose their home or decide to stay based on factors other than the proximity to services and access to employment [39].” 

Estonians and Latvians demonstrate high mobility rates either on a permanent basis or as part of a repeated pattern of circular migration. For many of the interviewees it was a conscious choice to remain in the place despite the limitations. Many others had decided to leave, however, due to the economic constraints. As Anton & Lawrence [reference39 p458] states “People who live in rural areas often choose to do so as they are drawn to the environment in these places.” The interviews were from a mix of people who had chosen to stay, return or move to the areas. During Soviet times it was actively encouraged to move to reduce place identity.

145-147. I do not understand this statement. Many researches have deeply investigated the landscape history and evolution, both at national and local scales. All the landscape history field as well as multitemporal analysis methodology-based researches, are focused on describing (and measuring!) the driving factors that have caused landscape construction and changes. Also the topic of people-place relationships have been deeply investigated, even with different methodologies, many studies focuses on landscape perception.

Argument accepted and sentence clarified. Now at line 178-180

180-182. Even if I do not directly know the two places, I am not sure that Obinitsa can be compared to Dagda. Even if both of them are located far from the main cities, the second one is a small town with more than 2000 inhabitants, while Obinitsa has only 135 inhabitants and is mainly made of sparse houses.

It is true that in terms of population the two areas cannot be directly compared but in other respects – remoteness, marginalisation, access to resources, etc they are comparable to a large degree. 

205-208. I am not sure that the answers given by the population can be representative of the real “place attachment”. The interviewed people range from 0,6% in Dagda to 2% in Tuja, except for Obinitsa where about the 28% of the population has been involved. In some places only 6 or 7 people have been interviewed. It is a big difference. Moreover, the use of snowball method can influence the sample, if people tend to suggest other people with similar background, opinions, habits,…

This point would be true if we were interested in obtaining a representative sample but since the work is qualitative, smaller numbers can suffice as long as specific safeguards against finding similar people through the eg snowball sampling technique are taken Our knowledge of the countries and locations enabled us to judge any undercurrents that each interview portrayed. In rural areas there is a degree of homogeneity within parishes and key representatives allowed us to deduce if we had captured the essence of the place or needed further investigations. Activists were instrumental in us obtaining the interviews or giving us key places to find other interviewees. Explanatory notes have been added. Lines: 882-897

319. “a criminal reputation for criminality” does not sound good. Please, correct the sentence.

Corrected to “a criminal reputation”

540. The meaning of the Figure 4 is not completely clear to me. On the right, please correct “culutural”.

Further clarifying text has been added to Figure 4 and culutural corrected.

Reviewer 2 Report

The article is interesting, especially the results that are clearly presented. But I have some doubts about the methodology and the aim of the research. According to the title, it should be about “place attachment, landscape planning and readiness to participate”. However, it seems to me that is only focused on the first topic. No information about the local landscape planning system and no information about the possible involvement of population are given. It seems to be a research on place attachment, but there is no sufficient information for an efficient landscape planning, since the main collected data refers to the feelings and the emotions of some citizens. In lines 596-599 authors state that “…is capable of revealing the values of the citizens planners need to for (please correct) sensitive planning”. I do not agree with this statement, the values identified talking to some (few) citizens are not enough for landscape or territorial planning. For an adequate landscape planning there is the need to deeply study the landscape structure and transformations, landscape perception, importance of different landscape features, needs and expectations of the local population.

Some examples of studies of the importance of public participation in landscape planning in different countries and frameworks:

Eiter, S.; Vik, M.L. Public participation in landscape planning: Effective methods for implementing the European Landscape Convention in Norway. Land Use Policy 2015, 44, 44–53

Dian, A.M.; Abdullah, N.C. Public participation in heritage sites conservation in Malaysia: issues and challenges. Procedia-Social and Behavioral Sciences 2013, 101, 248-255.

Bruns, D. Cultural Landscape: All That People Give Value to in Their Surroundings. In: Basic and Clinical Environmental Approaches in Landscape Planning. Shimizu H., Murayama A., Eds.; Springer, 2014. pp 3-13.

Paavola, J.; Hubacek, K. Ecosystem services, governance, and stakeholder participation: an introduction. Ecology and Society 2013, 18(4), 42.

Prieur, M.; Durousseau, S. Landscape and public participation. In: Landscape and Sustainable Development. Challenges of the European Landscape Convention. AA.VV.; Council of Europe Publishing: Strasbourg, France, 2006.

UNECE United Nations Economic Commission for Europe. Aarhus convention. 1998. www.unece.org/env/pp/treatytext.htm

Selman, P. Community participation in the planning and management of cultural landscapes. Journal of Environmental Planning and Management 2004, 47, 365–392

Stenseke, M. Local participation in cultural landscape maintenance: Lessons from Sweden. Land Use Policy 2009, 26, 214–223.

Ryan, R.L. The social landscape of planning: Integrating social and perceptual research with spatial planning information. Landscape and Urban Planning 2011, 100, 361–363.

Here some punctual observations:

54. Specify briefly the “various conventions and policies”. At least name the European Landscape Convention, for sure the most important in the last decades.

126. I would not use the word “choose”. Home is not always a choice, but in most of the cases it is simply the place where you are born or where you grew up.

145-147. I do not understand this statement. Many researches have deeply investigated the landscape history and evolution, both at national and local scales. All the landscape history field as well as multitemporal analysis methodology-based researches, are focused on describing (and measuring!) the driving factors that have caused landscape construction and changes. Also the topic of people-place relationships have been deeply investigated, even with different methodologies, many studies focuses on landscape perception.

180-182. Even if I do not directly know the two places, I am not sure that Obinitsa can be compared to Dagda. Even if both of them are located far from the main cities, the second one is a small town with more than 2000 inhabitants, while Obinitsa has only 135 inhabitants and is mainly made of sparse houses.

205-208. I am not sure that the answers given by the population can be representative of the real “place attachment”. The interviewed people range from 0,6% in Dagda to 2% in Tuja, except for Obinitsa where about the 28% of the population has been involved. In some places only 6 or 7 people have been interviewed. It is a big difference. Moreover, the use of snowball method can influence the sample, if people tend to suggest other people with similar background, opinions, habits,…

319. “a criminal reputation for criminality” does not sound good. Please, correct the sentence.

540. The meaning of the Figure 4 is not completely clear to me. On the right, please correct “culutural”.

Author Response

Comments and Suggestions for Authors

Dear authors, it has been a great pleasure to have been able to review your article. It has been very interesting and important for me to identify and diagnose aspects related to the social level for landscape planning, especially in that particular context. Really, as indicated by the European Commission of landscaping, public participation processes are key and their mechanisms must be improved. Congratulations. The only thing that can be improved is the presentation of the results, I understand that it can be complicated but it would make it easier to read and compare the fields of study.

Thank you for your kind words and encouragement. As you mentioned, presenting the results is complicated, especially as our aim was to ensure that the voices of the rural people were also represented. We have added further clarifications in the discussion, added a table and amalgamated two sections, which we hope improves the results. 

By other way, Could be possible any reference to similar studies in other countries?

Owing to a lack of space it was not possible to add similar studies from other countries, however, it would make an excellent topic for a collaborative paper in the future, where the similarities and differences could be discussed at greater length

Reviewer 3 Report

Dear authors, it has been a great pleasure to have been able to review your article. It has been very interesting and important for me to identify and diagnose aspects related to the social level for landscape planning, especially in that particular context. Really, as indicated by the European Commission of landscaping, public participation processes are key and their mechanisms must be improved. Congratulations. The only thing that can be improved is the presentation of the results, I understand that it can be complicated but it would make it easier to read and compare the fields of study.

By other way, Could be possible any reference to similar studies in other countries?

Best wishes.

Author Response

Comments and Suggestions for Authors

This paper has some technical merits in its investigation of place attachment, landscape planning and readiness to participate using a case study approach. The paper is theoretically well founded. The manuscript has good potential to attract readership but however does not currently meet the publication standards of Sustainability and needs some revisions.

The publication that this paper was sent to was “Land” and not “Sustainability”

The authors however need to articulate more clearly how the current study bridges the gaps between theory and practice. The aims and objectives of the research need to be stated at the outset of the paper.

Additions have been made throughout the paper to bridge the gap between place attachment and landscape-scale management. The aims and objectives are stated in lines 166-175

The authors need a literature review section. There are a number of recent work on place attachment e.g. Ramkissoon, Mavondo, & Uysal, 2018 (Journal of Sustainable Tourism); Jiang et al., 2017 (Journal of Hospitality Marketing & Management that may help to enhance the flow of arguments. Authors also need to review literature on community participation – Nunkoo et al. (2012) Annals of Tourism Research and  stakeholder engagement e.g. Sowamber et al. (2018); Nunkoo & Ramkissoon (2017) - Journal of Hospitality & Tourism Research

There is insufficient information listed here to ensure the correct papers are found. Additionally, the journals are focussed mainly on tourism, which was not the focus of this paper. A search of the Journal of Hospitality and Tourism Research did not find an article by Sowamber in any year or by Nunkoo and Ramkissoon in 2017.

The methodology is sound. The discussion of findings however needs to be adequately tied to literature. This section needs to be carefully revised and linked to relevant studies. Authors should clearly stress on the selling points on their paper. The conclusion needs to tie to the rest of the manuscript. Overall, this is an interesting manuscript and if carefully revised, it has potential to attract good readership. Thank you, and good luck with the revisions.

 Further additions of literature references have been added to the discussions. 

Round 2

Reviewer 2 Report

The answers given by the authors and the changes made to the paper have clarified the main doubts. The article is suitable for publishing.